# A Fast-Responding Electro-Activated Shape Memory Polymer Composite with Embedded 3D Interconnected Graphene Foam

**DOI:** 10.3390/mi13101589

**Published:** 2022-09-24

**Authors:** Yucheng Zhou, Jianxin Zhou, Jiasheng Rong, Cong Hu

**Affiliations:** State Key Laboratory of Mechanics and Control of Mechanical Structures, Key Laboratory for Intelligent Nano Materials and Devices of Ministry of Education, Institute of Nanoscience and College of Aerospace Engineering, Nanjing University of Aeronautics and Astronautics, Nanjing 210016, China

**Keywords:** three-dimensional interconnected graphene foam, shape memory polymers, electrothermal response, electro-activated

## Abstract

Shape memory polymers (SMPs) have gained increasing attention as intelligent morphing materials. However, due to the inherent electrical insulation and poor thermal conductivity of polymers, deformation and temperature control of SMPs usually require external heating devices, bringing about design inconveniences and fragility of interfaces. Herein, we report a shape memory composite that integrates reliable temperature and shape control functions into the interior. The composite is comprised of resin-based SMP and three-dimensional interconnected graphene foam (3DGF), exhibiting a high recovery rate and thermal/electrical conductivity. With only 0.26 wt% of graphene foam, the composite can improve electrical conductivity by 15 orders of magnitude, thermal conductivity by 180%, tensile strength by 64.8%, and shape recovery speed by 154%. Using a very simple Joule heating scheme, decimeter-sized samples of the composite deformed to their preset shapes in less than 10 s.

## 1. Introduction

As a shape memory material, shape memory polymers (SMPs) have the advantages of low density, large deformation, low cost, and good biodegradability compared to shape memory alloys and shape memory ceramics [1,2]. In recent years, with the development of flexible electronics and soft robots, the characteristics of materials such as good softness, high flexibility, and large extensibility have become highly desirable, which coincides with the properties of SMPs [3,4]. The deformation of SMPs is usually thermally actuated and modulated by their heating temperature, but like other polymers, SMPs have very low electrical/thermal conductivity and usually require an external heating device to control their temperature and deformation state, which always increases the design complexity, reduces the deformability and brings unreliability, thus greatly hindering their practical applications [5,6,7,8,9,10]. To improve the manipulability of SMPs, various conductive fillers, including carbon black [11], carbon nanotubes [12], silver nanoparticles [13], and copper pastes [14], have been investigated and added to the SMPs to improve their electrical conductivity. 

A good conductive filler needs to meet three conditions: (1) to form a uniform dispersion in the polymer, (2) to build a homogeneous three-dimensional conductive network, and (3) to constitute effective connections between the microstructures of the filler and the polymer molecules [15]. Graphene is considered to be a promising conductive filler owing to its remarkable intrinsic features, such as high electron mobility (>2.5 × 10^5^ cm^2^V^−1^s^−1^), high thermal conductivity (3000–5000 W mK^−^^1^), and high tensile strength (130 GPa) [10,16]. Various graphene materials, including reduced graphene oxides (RGO) [17,18,19], ultrasonically dispersed graphene solutions [20], and graphene nanoflakes [21] have been used as additives. However, most of the graphene used is in the form of powder, and powdered fillers tend to agglomerate and are hard to disperse uniformly, making it difficult to establish three-dimensional conducting networks at a low filling rate [12,22], while a higher filling rate often causes the degradation of mechanical and shape recovery performance of the SMP [23]. In addition, the powdered fillers usually adsorb small molecules on their surfaces, which significantly reduces the binding force between the fillers and the SMP molecular chains, thus the conductivity of composites usually seriously decreases after several deformation cycles [24].

Compared to graphene powders, graphene foams (GFs) have higher conductivity [25,26,27], more uniform distribution, and higher porosity [28]. However, seldom have studies been reported on electro-activated SMPs embedding chemical vapor-deposited graphene foams (CVD-GFs). Idowu et al. prepared an SMP composite using CVD-GFs, but their GFs were just simple commercial products, and the CVD growth process and microstructures were not optimized for electro-activated SMPs [29]. Here, we prepare graphene foam using an optimized CVD method, and we employ this three-dimensional interconnected graphene foam (3DGF) into an E51-based epoxy resin shape memory polymer (E-SMP) to form a graphene foam embedded shape memory polymer (GF-SMP). The 3DGF can efficiently functionalize the SMP; with just 0.26 wt% of 3DGF, the electrical conductivity of the SMP improved by 15 orders of magnitude, the thermal conductivity by 180%, the tensile strength by 64.8%, and the shape recovery speed by 154%. With a very easy Joule heating scheme, an electro-activated GF-SMP sample can be restored to its preset shape in less than 10 s. The applications of the GF-SMP in simple deformable structures and a wing-like model are also demonstrated.

## 2. Materials and Methods

The materials used were as follows: foam nickel (Suzhou, China, Kunshan Guangjiayuan Materials Co., Ltd.), FeCl_3_ (Shanghai, China, Aladdin Chemical Reagent Co., Ltd.), epoxy resin E51(Nantong, China, Nantong Xingchen synthetic material Co., Ltd.), polysulfide rubber (Wuhan, China, Wuhan kanos Technology Co., Ltd.), aminoethyl piperazine (Shanghai, China, Aladdin Chemical Reagent Co., Ltd.), butyl glycidyl ether (Changzhou, China, Changzhou Runxiang Co., Ltd.).

### 2.1. Preparation of 3DGF

As shown in Figure 1a, the preparation of 3DGF was similar to the chemical vapor deposition (CVD) method that we reported previously [30], but growth conditions were adjusted to achieve better shape memory performance. Generally, when the carbon source, methane, undergoes chemical decomposition at a high temperature of 1000 °C, the decomposed carbon atoms will penetrate and diffuse into the bulk phase of the nickel foam, and the dissolved carbon will segregate on the surface of the nickel foam due to supersaturation during cooling, forming a graphene layer. After CVD growth, the nickel foam skeleton was etched off with a solution of ferric chloride, and then the samples were freeze-dried to obtain dry, three-dimensional interconnected graphene foams.

Figure 1b shows a photograph of a 3DGF with dimensions of 20 × 98 × 1 mm^3^, which had a porous structure and a density of 4.68 mg·cm^−3^, 3.6 times that of air, indicating its aerogel-like, ultralight, nature. The corresponding Raman spectrum (532 nm laser) is shown in Figure 1c; the characteristic 1580 cm^−1^ (G) and 2705 cm^−1^ (2D) peaks confirm its graphene structure. For most of the randomly selected measurement points, the intensity of the 2D peak is notably higher than that of the G peak, indicating that the main form of graphene existing in the GF is a monolayer structure [31].

### 2.2. Synthesis of E-SMP

The E-SMP precursor was prepared using epoxy resin E51 as the main material, aminoethyl piperazine as the curing agent, polysulfide rubber as the toughening agent, and butyl glycidyl ether as the diluent. All raw materials were mixed in a beaker in the ratio of main material (100): curing agent (24): toughening agent (35): diluent (20), and stirred thoroughly to obtain a clear solution of SMP precursor.

### 2.3. Preparation of GF-SMP

Taking an example of preparing a 110 × 20 × 1.8 mm^3^ sample: 4.44 g of the clarified solution of E-SMP precursor was prepared in advance, and the precursor was slowly dripped into the glass mold containing the 3DGF sample. After the 3DGF was completely covered by the precursor solution, the whole device was put into a vacuum chamber, brought to 10 Pa, and maintained for 5 min. Then, the device was placed on a heating table at 80 °C for 4 h, and the GF-SMP was obtained after demolding. Figure 1d shows a photograph of a resulting GF-SMP sample.

### 2.4. Characterization

A scanning electron microscope (SEM, Zeiss Evo 18) was used to characterize the microstructure of the graphene foam and composites, and a digital camera (Nikon D5600) was used to capture the optical images. A Horiba LabRAM HR Evo Raman scattering spectroscopy system was used to analyze the 3DGF samples. The surface temperature distributions of the composites were recorded during the recovery processes using a thermal infrared imager (FOC 220s). The electrical signals were measured using a digital multimeter (Keithley, DMM7510), and a sourcemeter (Keithley 2450) provided the DC power.

The stress–strain curves of the E-SMP and GF-SMP samples were determined using a tensile testing machine (Youhong UH6104) with a sample size of 90 × 20 × 1.8 mm^3^, and each group of experiments was repeated at least five times. A dynamic mechanical analyzer (TA, Discovery DMA 850) was used to control the heating rate via programming and applying dynamic load to the samples, so as to determine thermodynamic parameters such as storage modulus, loss modulus, and loss angle. A xenon lamp thermal conductivity meter (TA, DXF 900) was used to measure the thermal conductivity of the E-SMP and GF-SMP.

## 3. Results and Discussion

### 3.1. Microstructure

SEM morphology results of the graphene foam structure (Figure 2a) show that the graphene was convoluted into irregular three-dimensional tubular structures with voids of several hundred microns; the tubes had diameters of 50–100 microns (the histogram is shown in the inset of Figure 2a), and the overall network maintained good three-dimensional connectivity and homogeneity. The magnified micrograph (Figure 2b) shows a cross-section of a broken graphene tube, indicating that these tubes were very thin-walled hollow structures. To investigate the compounding of the polymer with graphene foam, we observed the cross-sectional structure of a GF-SMP sample after stretching and fracturing it (Figure 2c). The smooth surface presented in Figure 2d is the fracture surface of the resin polymer, and the rough surface covered with wrinkles is the graphene surface exposed after the graphene tubes were pulled off. Note that the polymer filled not only the space between the graphene tubes in the foam, but also the interior of the graphene tubes (Figure 2e,f), which proves that our preparation method caused the polymer to bond well with the graphene network.

### 3.2. Thermal Conductivity

Constructing a high thermal conductivity graphene network into a low thermal conductivity polymer is an effective way to improve the thermal properties of composites. As an example, we compared the thermal conductivity of the E-SMP samples unfilled with graphene foams to that of the GF-SMP samples filled with graphene foams under identical measurement conditions. Figure 3a shows that the average thermal conductivity of the E-SMP samples (green bars) was 0.69 W ± 0.03 Wm^−1^K^−1^ and that of the GF-SMP samples (purple bars) was 1.99 ± 0.05 Wm^−1^K^−^^1^. The thermal conductivity of the composite increased by 180% at a filler ratio of only 0.26 wt%, indicating that 3DGF is a very efficient thermal-conductivity enhancing filler, acting outstandingly at a very low content. In contrast, conventional fillers (carbon black, metal nanoparticles, etc.) need to be filled at a rather higher percentage (e.g., >2 or even >10 wt%) to achieve results similar to those of 3DGF [32,33,34]. This advantage of 3DGF is due not only to the high thermal conductivity of CVD graphene itself, but also to the uniformly connected 3D graphene network that is very favorable for phonon and thermal transport.

To further investigate the thermal conductivity performance, E-SMP and GF-SMP samples of the same size (100 × 20 × 1.8 mm^3^) were heated from the lower surface using a heating film (heating power 20 W, ambient temperature 36 °C), and the temperature of the upper surface of the samples was recorded using a thermocouple. The recorded temperature of the GF-SMP sample (Figure 3b, red line) increased significantly faster than that of the E-SMP sample (blue line). After 10 s of heating, the temperature of the GF-SMP sample increased by 9.4 °C, while that of the E-SMP sample increased by only 3.9 °C. The infrared photos in the left insets of Figure 3b also show similar results. All these data prove that the 3DGF networks substantially increased the heat transfer performance of the composites.

### 3.3. Electrical Conductivity

Resin-based SMPs are generally insulating materials, e.g., the electrical conductivity of our E-SMP samples was less than 1 × 10^−13^ Sm^−1^. After filling with 3DGF, the GF-SMP achieved a conductivity of approximately 133 Sm^−1^, which was 15 orders of magnitude higher than that of the E-SMP. A current–voltage (I–V) curve of a GF-SMP sample (100 × 20 × 1.8 mm^3^) is displayed in Figure 3c, showing good electrical conductivity and ohmic contact. The powdered, discrete conductive fillers (CB, CNT, RGO, etc.) often require high filler ratios (>5 or even >10 wt%) to achieve a conductivity of approximately 100 Sm^−1^ [17,35,36,37,38,39,40,41,42,43,44], whereas the graphene networks require only very low filler ratios to achieve this value [45]. Figure 3d shows a comparison of the contents of common conductive fillers and the corresponding conductivity of the composites, illustrating the great advantage of 3DGF.

Carbon materials are the most commonly used materials for Joule heat generation, and the conductive graphene foam embedded in the polymer forms a homogeneous 3D heating network. We investigated the Joule heat generation behavior of the GF-SMP samples at different voltages (Figure 3e). The experiments were conducted in an open space with an ambient temperature of 22 °C, and during each experiment, the driving voltage was constant. When a voltage of 10 V was applied, the temperature of the sample increased by 5.9 °C in 10 s, and as the heating time continued, the rate of temperature increase slowed down; the surface temperature reached a maximum of 43.1 °C after 52 s. With a voltage of 20 V, the temperature increased by 29.2 °C in 10 s and reached a maximum of 119.1 °C after 50 s. This suggests that the GF-SMP can be easily electrically heated to control its temperature, and thus its shape recovery process.

### 3.4. Mechanical Properties

We also investigated the effect of 3DGF on the mechanical properties of the composites. The stress–strain curves of GF-SMP and E-SMP samples are shown in Figure 4a, where the test samples were both 90 × 20 × 1.8 mm^3^ in size, and each set of tests was repeated at least five times. The maximum stress of the E-SMP samples was 14.9 MPa, while that of the GF-SMP samples was 23.0 MPa. The introduction of graphene foam increased the tensile strength of the composite by 64.8%, which was consistent with the previously reported results that graphene networks can notably increase the mechanical strength of flexible materials [46]. Other powdered fillers [47,48] can also enhance the strength of the composites, but rather higher filling content and good control of homogeneity are required. A very small amount of 3DGF can effectively enhance the mechanical properties, which may come from two aspects: the ability of the planar honeycomb lattice structure of graphene to be better bonded with the polymer molecular chains, and the 3D foam structure that ensures homogeneity from microscopic to macroscopic scale.

The graphene foam has only a very slight effect on the thermomechanical properties of the SMP. In Figure 4b, there is almost no difference between the energy storage modulus of E-SMP and GF-SMP, indicating that the elasticity (or stiffness) of the material is nearly unchanged. In regard to the loss factor (Figure 4c), the peak positions of the E-SMP and the GF-SMP are very close to each other, corresponding to a transition temperature of 47.0 °C for E-SMP and 47.9 °C for GF-SMP. However, the smaller peak width of tan δ for GF-SMP implies a faster shape recovery speed [49].

### 3.5. Shape Memory Performance

To evaluate the electro-activated shape memory performance, we compared the shape recovery process of the GF-SMP and E-SMP samples. The GF-SMP sample was brushed with silver paste at both ends to obtain two well-conducting electrodes. A polyimide–constantan thin film heater was attached below the E-SMP sample (Figure 5a); this consisted of polyimide film as the substrate and constantan foil as the resistive element; and the resistance of the film heater was controlled to be the same as that of the GF-SMP sample. The temporary shape of each sample was set as a U-shape at a temperature of 80 °C, then the U-shape was fixed at room temperature (Figure 5b). The shape fixity rate (*R*_f_) is defined as *θ*_f_/180°×100%, in which *θ*_f_ is the bending angle (°) for the U-shape. The *R*_f_ of the E-SMP was 98.6% and that of GF-SMP was 99.7%. The shape recovery ratio (*R*_r_) is defined as (*θ*_max_−*θ*_t_)/*θ*_max_×100%, in which *θ*_t_ is the bending angle at the time *t*, and *θ*_max_ is the pre-bending angle of the sample. The Joule-heat driven shape recovery processes are displayed in Figure 5c, with a side view photo to show the electrode and the wire in the inset. Obviously, the shape recovery rate of the GF-SMP sample was remarkably faster than that of E-SMP for the same sample size and the same heating power. Figure 5d shows the shape recovery rate versus time for GF-SMP and E-SMP samples. At a voltage of 20 V, an E-SMP sample required 61 s for full shape recovery, while a GF-SMP sample required only 24 s, showing a 154% improvement in speed.

### 3.6. Exhibitions

GF-SMP material can be straightforwardly designed and machined into electro-driven deformable structures. Figure 6a shows the fast recovery process of a 10 × 2 cm^2^ GF-SMP sample, which recovered from a U-shape to a flat shape within 10 s. Figure 6b illustrates the unfolding processes of a joint-like structure. The structure consists of two layers of material, the upper blue layer is the 3D printed component, and the lower layer is GF-SMP, providing deformation force for structural deformation. When the electrical current was passed through the GF-SMP layer, the joint-like structure changed from a temporary L-shape to a flat shape.

Figure 6c demonstrates an example of in-situ pre-programming with a designed prestressed component. The structure consists of three layers, with blue plates in the upper layer, a pre-stretched 3D printed elastomer in the middle layer, and a GF-SMP material in the bottom layer. After the power was switched on, the GF-SMP gradually changed to a soft rubber state until the pre-stretched elastomer was able to pull it from a flat shape into an L-shape. Then when the power was turned off, the structure rapidly cooled, and the L-shape was fixed. 

Figure 6d shows a rather larger deformable structure: two 3D-printed wing-like components connected by a GF-SMP component, with an overall length of 52 cm. This structure also quickly transformed under an electric current from a folded shape to an unfolded planar shape within 10 s.

Due to their limited thermal and electrical conductivities, the applications for SMPs are relevantly limited. However, our simple examples above show that deformable structures with good machinability and controllability can be very conveniently achieved using GF-SMP.

## 4. Conclusions

In summary, we have developed electro-activated GF-SMP material capable of versatile deformable structure applications. Composed of shape memory polymer and 3D interconnected graphene foam embedded within, the GF-SMP exhibits an improvement in electrical conductivity by 15 orders of magnitude, thermal conductivity by 180%, tensile strength by 64.8%, and shape recovery speed by 154%. The combination of easy processing and controllability of the GF-SMP allowed us to conveniently demonstrate rapid shape restorations in less than 10 s. The electro-triggering and rapid response of GF-SMP pave the way for practical applications in deployable structures, morphing wings, and soft robots.

## Figures and Tables

**Figure 1 micromachines-13-01589-f001:**
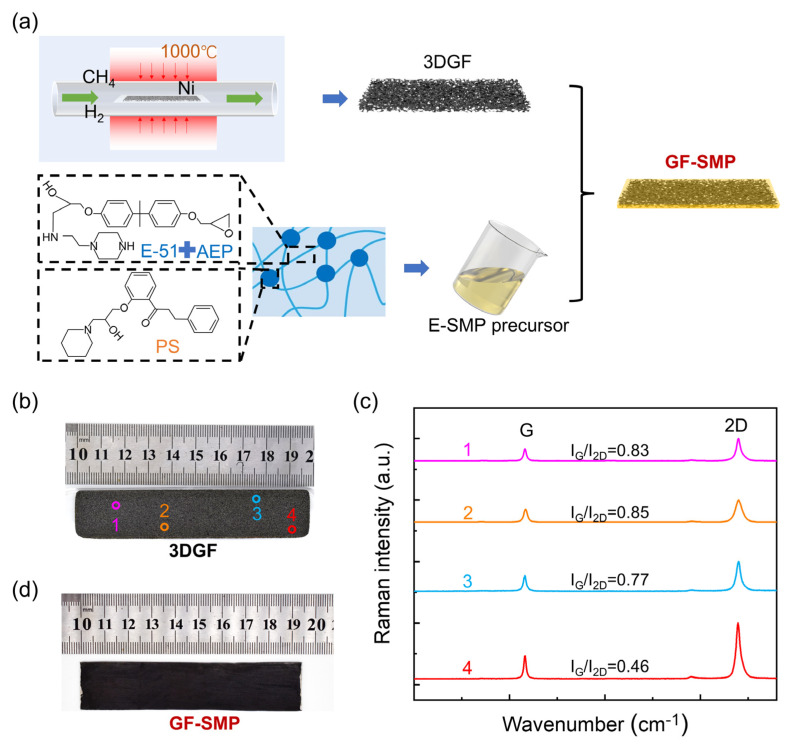
(**a**) Synthetic procedure for GF-SMP, (**b**) Image of a graphene foam sample, (**c**) Raman spectra of the graphene foam, (**d**) Image of a GF-SMP sample.

**Figure 2 micromachines-13-01589-f002:**
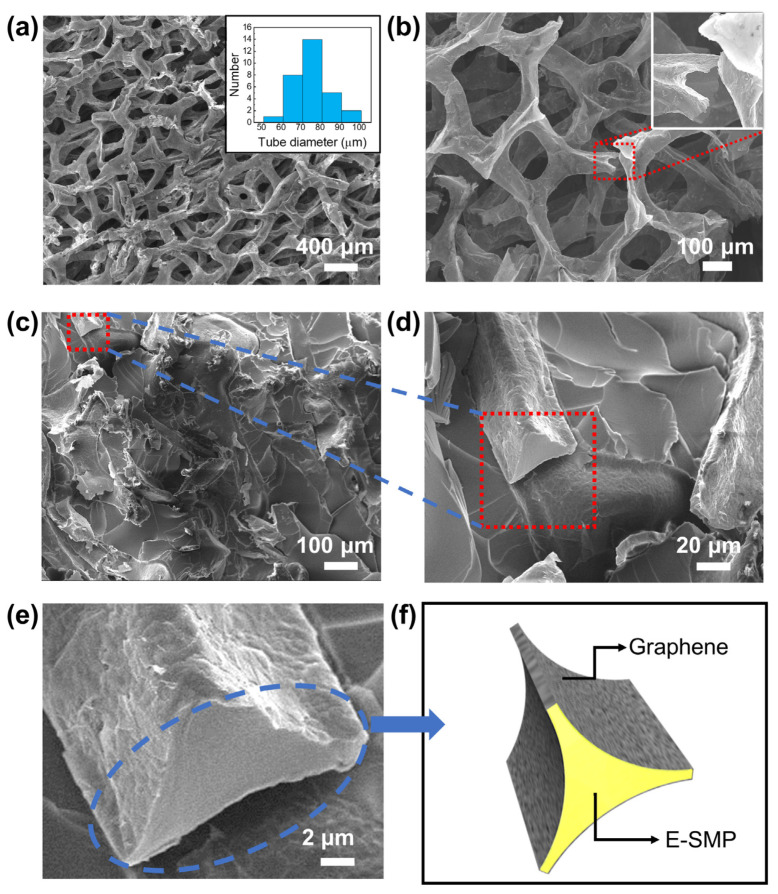
SEM images of (**a**) a graphene foam sample with histogram of tube diameter, (**b**) the cross-section of the foam with thin-walled hollow structures, (**c**–**e**) the cross-section of a GF-SMP sample after tensile fracture, and (**f**) schematic diagram of (**e**).

**Figure 3 micromachines-13-01589-f003:**
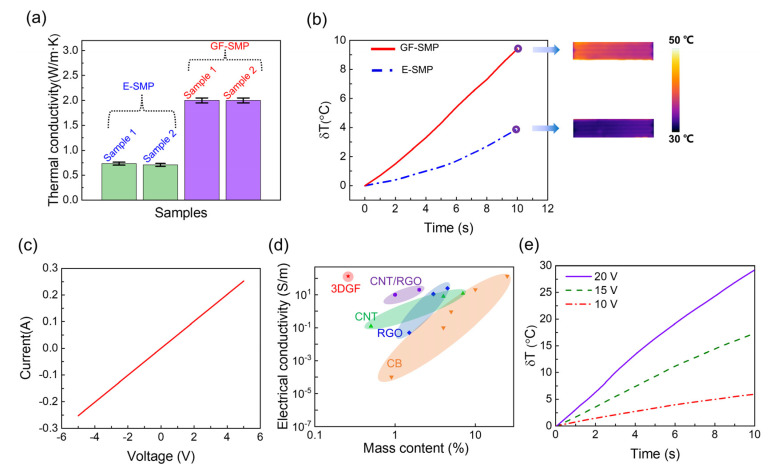
(**a**) Comparison of thermal conductivity of GF-SMP samples and E-SMP samples, (**b**) Comparison of temperature evolution of GF-SMP and E-SMP samples, both heated by a heating film, (**c**) I–V curve of a GF-SMP sample, (**d**) Comparison of electrical conductivity of SMP versus the mass percentage of different conductive fillers: carbon black (CB) [35,36,37,38], reduced graphene oxide (RGO) [39,40,41], carbon nanotube (CNT) [42,43], and reduced graphene oxide/carbon nanotube (CNT/RGO) [17,44], (**e**) Temporal evolution of temperature of a GF-SMP sample with time under different voltages.

**Figure 4 micromachines-13-01589-f004:**
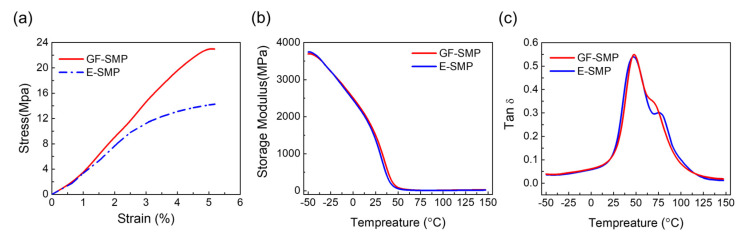
Comparison of mechanical properties of GF-SMP and E-SMP. (**a**) Comparison of stress–strain curves for GF-SMP and E-SMP samples, (**b**) Storage modulus plotted against the temperature for GF-SMP and E-SMP samples, (**c**) Loss factor plotted against temperature for GF-SMP and E-SMP samples.

**Figure 5 micromachines-13-01589-f005:**
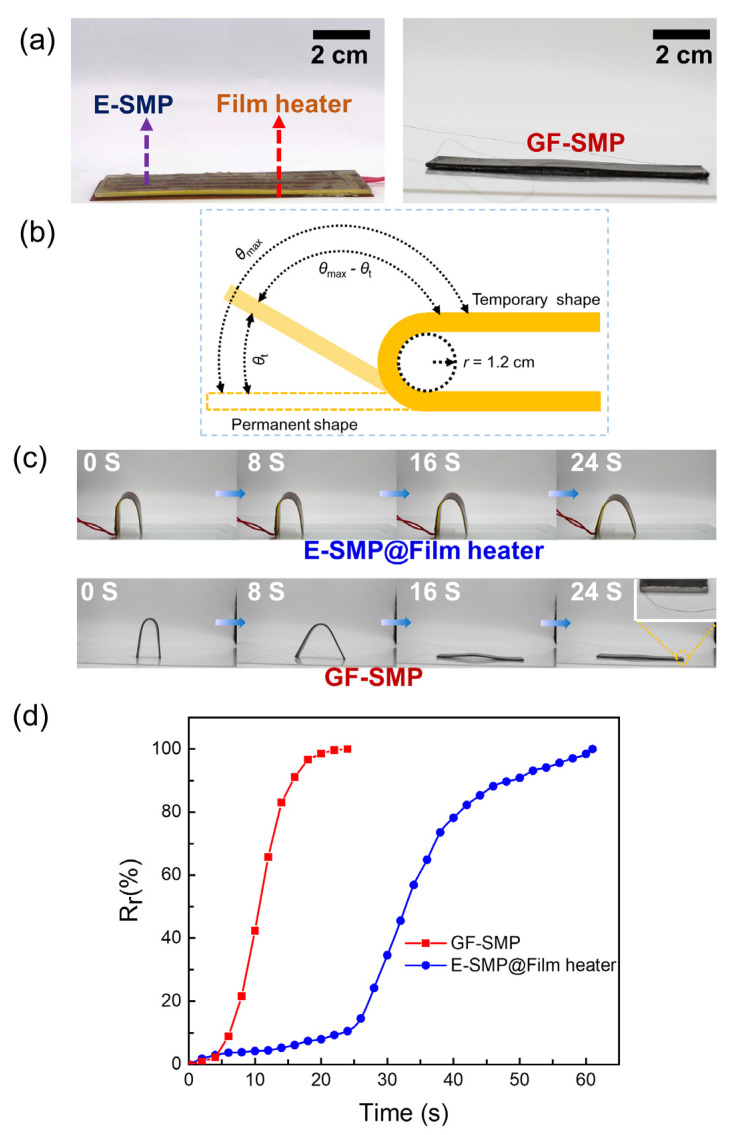
Comparison of shape memory properties of GF-SMP and E-SMP samples. (**a**) Photographs of an E-SMP sample with a thin film heater wrapped underneath (E-SMP@Film-heater, left) and a GF-SMP sample (right), (**b**) Shape memory model, (**c**) Electro-triggered shape recovery process of the E-SMP@Film-heater and the GF-SMP sample (inset: a side view photo to show the electrode and the wire), (**d**) Time dependent shape recovery rate for the E-SMP@Film heater and GF-SMP sample.

**Figure 6 micromachines-13-01589-f006:**
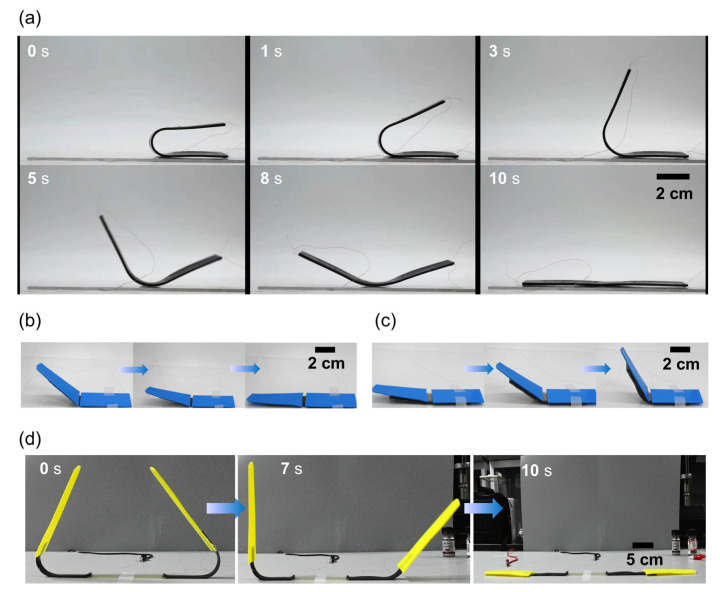
The shape recovery processes of GF-SMP structures. (**a**) Shape recovery of a 10 cm-long GF-SMP sample, (**b**) Unfolding of a GF-SMP joint structure, (**c**) Folding of a GF-SMP joint structure, (**d**) Unfolding of a 52 cm-long wing-like GF-SMP structure.

## Data Availability

Data is available upon request from the corresponding author.

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
