# Peer review of "A Fast-Responding Electro-Activated Shape Memory Polymer Composite with Embedded 3D Interconnected Graphene Foam"

_micromachines, 2022, doi:10.3390/mi13101589_

Round 1
Reviewer 1 Report
The authors of the manuscript with the title: "A Fast-Responding Electro-Activated Shape Memory Polymer Composite with Embedded 3D Interconnected Graphene Foam" investigate an epoxy resin SMP/3D graphene foam-based composite with improved electric/thermal conductivity and mechanical strength compared to neat polymer. Nevertheless, the authors did not cite the recent literature about graphene-SMP composites. There are number of reports about this topic. Some of them listed below
1. Appl Mater Today. 2019;15: 185 - 91.
2. Polymer. 2011;52(7):1603–11.
3. ACS Appl Mater Interf. 2016;8(33):21691–99.
4. Micromachines. 2016;7(8):145.
5. Thesis by Adeyinka Idowu Florida International University DOI: 10.25148/etd.FIDC007844
Based on the published work, a major revision of the introduction part is required. Authors should highlight the innovation in this work compared to the already published work. However, I suggest to consider the manuscript for publication after a major revision. Please see my comments below.
Question1: Authors should provide the difference between E-SMP-1 and E-SMP-2 or GF-SMP-1 and GF-SMP-2. These abbreviations are not introduced in the main text.
Question 2: The authors claim that SMP filled not only the space between the graphene
tubes in the foam but also the interior of the graphene tubes. A cross section of the hollow graphene tube before introduction of SMP resin would be helpful to prove the claim.
Questions 3: During electric heating of GF-SMP sample as shown in Figure 3e, was any plateau temperature achieved??? As the temperature rise of upto 10 s is shown. Was it a controlled heating or uncontrolled rise in temperature continued? Authors should add the explanation in the manuscript. Or graphs with plateau region should be shown.
Questions 4: What was the target temperature to observe the shape- recovery in SMP? The transition temperature of the SMP is not mentioned in the whole manuscript. If it is below 50 °C, why a SMP with such a low transition temperature was selected??
Question 5: The programming step of the GF-SMP is not explained in the manuscript. How was shape-fixity ratio ? and which temperature was used to fix the temporary programmed shape.
Question 6: In figure 5b, why polyimide paste was selected for E-SMP, as PI is an electrical insulator. The authors should highlight, how PI was made conductive???
Question 7: The mechanism of Figure 6c is not clearly explained. The authors should explain clearly, why it bends to L shaped without any pre-programming and what is the role of pre-stretched 3D printed part.
Question 8: In Figure 5b, the current source is not shown clearly for the Joule heating of GF-SMP. The authors should modify the pictures and clearly show where the electric wires were attached to two electrodes on both surfaces of the sample.

Reviewer 2 Report
Comments to the Authors
1. In the introduction section authors should discuss more about graphene and its related composite. For this discussion authors can use the following articles
a) doi.org/10.1016/j.surfin.2018.08.004
b) doi.org/10.1016/j.compositesb.2017.09.046
2. Authors should provide a histogram of tube diameter.
3. An error bar should be provide in Figure 3 a.
4. Few grammatical errors are present in the manuscript. Authors should carefully checked it.
Round 2
Reviewer 1 Report
There is a major improvement in the quality of the manuscript and is suitable for publication now.